# Effect of Ginger on Inflammatory Diseases

**DOI:** 10.3390/molecules27217223

**Published:** 2022-10-25

**Authors:** Pura Ballester, Begoña Cerdá, Raúl Arcusa, Javier Marhuenda, Karen Yamedjeu, Pilar Zafrilla

**Affiliations:** Nutrition, Oxidative Stress and Bioavailability Group, Degree in Pharmacy, Faculty of Health Sciences, Catholic University of San Antonio de Murcia, 30107 Murcia, Spain

**Keywords:** inflammatory diseases, ginger, bioactive compounds

## Abstract

Ulcerative colitis, Crohn’s disease, rheumatoid arthritis, psoriasis, and lupus erythematosus are some of common inflammatory diseases. These affections are highly disabling and share signals such as inflammatory sequences and immune dysregulation. The use of foods with anti-inflammatory properties such as ginger (*Zingiber officinale Roscoe*) could improve the quality of life of these patients. Ginger is a plant widely used and known by its bioactive compounds. There is enough evidence to prove that ginger possesses multiple biological activities, especially antioxidant and anti-inflammatory capacities. In this review, we summarize the current knowledge about the bioactive compounds of ginger and their role in the inflammatory process and its signaling pathways. We can conclude that the compounds 6-shoagol, zingerone, and 8-shoagol display promising results in human and animal models, reducing some of the main symptoms of some inflammatory diseases such as arthritis. For lupus, 6-gingerol demonstrated a protective attenuating neutrophil extracellular trap release in response to phosphodiesterase inhibition. Ginger decreases NF-kβ in psoriasis, and its short-term administration may be an alternative coadjuvant treatment. Ginger may exert a function of supplementation and protection against cancer. Furthermore, when receiving chemotherapy, ginger may reduce some symptoms of treatment (e.g., nausea).

## 1. Introduction

Ginger (*Zingiber officinale Roscoe*) is a member of the *Zingiberaceae* family of plants [1]. Some studies have proven that ginger is the most-used herbal drug in many countries [2]. Scientific evidence supports the beneficial properties of ginger, including antioxidant and anti-inflammatory capacities; in contrast, a more specific and less-studied bioactivity is the possible neuroprotective effect of ginger [3]. *Zingiber officinale* is among the medicinal plants with beneficial health effects that has been widely used in pharmaceutical products and food. Its crude extract is known for its pharmacological effects [4]. Ginger rhizome is commonly added to food as a spice or taken as a dietary supplement, and has been widely used in traditional medicine throughout history [5].

One of ginger’s main uses is to treat urinary tract inflammatory problems [6]. Furthermore, its anti-inflammatory properties, due to immune response modulation during the cellular phase, have been described. Another highlighted ability of this herbal extract are its antinociceptive effects induced by acetic acid [7]. Its bioactive compounds have an analgesic and anti-inflammatory effect by inhibiting COX2 and LOX pathways, therefore preventing arachidonic acid metabolism [8]. The effect of ginger has been shown to be similar to the NSAIDs family; however, it does not have a negative effect on stomach mucosa (see Figure 1).

We know that ginger does not affect mucosa, because a raise of mucosal prostaglandins synthesis has been measured after ginger intake, as it does not act as an inhibitor of COX1 [9]. Furthermore, a study of a less-than-two-week intervention with oral ginger supplements in osteoarthritis patients demonstrated ginger’s effectiveness as a pain reliever and anti-inflammatory compound, assessing muscle pain and plasma PGE2 levels, confirming its specificity to COX2 enzyme [10]. In recent years, ginger has also been found to possess biological activities, such as antimicrobial, antioxidant, and anti-allergic activities, as well as helping to prevent cancer (e.g., improvement in the expression level of markers for colorectal cancer risk). In this sense, numerous studies have demonstrated that ginger is capable of potentially preventing cardiovascular diseases, associated pathologies that act as cardiovascular diseases risk factors (diabetes, obesity, and metabolic syndrome), chemotherapy-induced emesis and nausea, arthritis, gastric dysfunction, pain, respiratory disorders, and neurodegenerative diseases. Ginger extract also showed antioxidant effects in human chondrocyte cells, with oxidative stress mediated by interleukin-1β (IL-1β). It stimulated the expression of several antioxidant enzymes and reduced the generation of ROS and lipid peroxidation. Additionally, ginger extract could reduce the production of ROS in human fibrosarcoma cells with H_2_O_2_-induced oxidative stress [11].

Some of the most common inflammatory diseases among the population are ulcerative colitis, Crohn’s disease, rheumatoid arthritis, psoriasis, and lupus. They are diseases with different characteristics and with different organ involvement, but in all of them there is an inflammatory and oxidative process which leads to a loss of immune regulation.

## 2. Bioactive Compounds in Ginger

Ginger is composed of multiple bioactive compounds that contribute to its recognized biological activities. Ginger has been identified as having a multitude of bioactive compounds, including phenolic compounds, terpenes, lipids, and carbohydrates. Hence, its pharmacological effects are largely attributed to phenolic compounds and terpenes [12,13].

Of the 400 types of compounds present in ginger, four phenolic compounds are mainly responsible for its biological effects (Table 1): gingerols, shogaols, paradols, and zingerone; overall, in vitro and in vivo studies have demonstrated their strong anti-inflammatory and antioxidant activity [11,14]. Figure 2 summarizes the structure and properties of the four main phenolic compounds of ginger. In fresh ginger, gingerols such as 6-gingerol, 8-gingerol, and 10-gingerol are the major polyphenols. With heat treatment or long-time storage, gingerols can be transformed into corresponding shogaols. After hydrogenation, shogaols can be transformed into paradols. There are also many other phenolic compounds in ginger, such as quercetin, zingerone, gingerenone-A, and polyphenols; all have showed high antioxidant activity [3].

At the present time, pure 6-gingerol, the major component in ginger rhizomes, can be obtained by total synthesis [15] and has shown several interesting pharmacological and physiological activities, such as anti-inflammatory, analgesic, and cardiotonic effects [2]. By dehydration and after long storage, this compound is converted into 6-shogaol, which is more stable and has greater pharmacological effects than its precursor 6-gingerol [16,17]. 6-shogaol is converted to 6-paradol by bacterial metabolism, and both possess similar anti-inflammatory and antioxidant properties [5,17]. Antioxidant, antitumoral, antilipemic, antibacterial, and anti-inflammatory actions are attributed to zingerone, and it is synthesized by reverse aldolization of gingerols when heating fresh ginger [17,18]. Furthermore, there are also other phenolic compounds related to gingerol (8-gingerol, 10-gingerol, 12-gingerol) and shogaol (1-dehydrogingerdione, 6-gingerdione, 10-gingerdione) that actively play a pharmacological role [13].

Ginger-derived terpenes (α-zingiberene, camphene, α-curcumene, β-sesquiphellandrene, α-farnesene, β-bisabolene, α-piene) [19] are known to avoid inflammatory processes and bacterial growth, have an antioxidant effect, help to prevent high blood sugar levels, act as painkillers or protectors of gastric tissue, and exert neuroprotective and anticarcinogenic properties [12].

## 3. Antioxidant and Anti-Inflammatory Properties

Scientific evidence supports the beneficial properties of ginger, including antioxidant and anti-inflammatory capacities; in contrast, a specific and less-studied bioactivity is the possible neuroprotective effect of ginger [3]. Some of these properties have been documented in a systematic review, where sufficient evidence was provided about ginger-activated improvements of oxidative stress parameters [18].

Oxidative stress takes place in the body when the mechanisms of antioxidation are not working, so a loss of equilibrium between the production and elimination of reactive oxygen species (ROS) occurs. Some of the effects described are due to the activation of the Nrf2 signaling pathway [20]. After that, ROS accumulation generates cellular damage through lipid peroxidation [21]. Free radicals can trigger a chronic inflammation status in the body that produces tissue destruction. However, ginger also promotes a rescue mechanism through the enzyme paraoxonase-1, avoiding lipidic oxidation (LDL) [22]. Ginger inhibits lipid peroxidation through its antioxidant effect. 6-gingerol increases Beclin1 expression to promote autophagy in endothelial cells and inhibits PI3K/AKT/mTOR pathway signaling without affecting cell cycle [15].

Previous studies [23] in animal models have described that ginger supplementation may protect against oxidative stress, increasing catalase and SOD activity [24]; some clinical trials documented that after a ginger administration, SOD, catalase, glutathione peroxidase, and blood glutathione were significantly augmented, while the oxidative stress markers MDA and nitric oxide were significantly diminished. Ginger has a beneficial effect in those conditions where an increased ROS production is described [25], together with a lipid peroxidation and tissue damage [26], mediated by inflammatory cytokines, especially TNF-α [27].

Several ginger bioactive compounds, such as 6-gingerol, 8-gingerol, 10-gingerol, and 6-shogaol, exhibit antioxidant activity. The highest antioxidant activity in vitro is 6-gingerol, followed by 6-shogaol [28]. 6-gingerol has been shown to be capable of inhibiting xanthine oxidase, an enzyme that catalyzes the oxidation of hypoxanthine to xanthine and of xanthine to uric acid in the last stage of purine metabolic degradation with the production of reactive oxygen species [29]. In addition, it has been proven that this compound is capable of increasing superoxide dismutase and catalase activity, two antioxidant enzymes [30]. In animal models, 6-gingerol has proven to regulate lipogenesis, fatty acid oxidation, mitochondrial dysfunction, and oxidative stress of aging rats. In addition, it increases the activity of the antioxidant enzyme superoxide dismutase (SOD) and decreases the levels of malondialdehyde (MDA), a marker of lipid peroxidation, in a concentration-dependent manner [31].

At this point, it needs to also be pointed out that the overproduction of ROS in the human body is a cause of many diseases. Theoretically, ginger-plant-based medicine, with its antioxidant effects, could be beneficial. However, several factors, such as health conditions, individual differences, lifestyle, other dietary factors, and the dosage, solubility, and oral intake of antioxidants could affect the bioaccessibility and bioavailability of antioxidants, leading to low blood concentrations overall, which probably could explain why most antioxidants do not work in the real world [11].

Several clinical trials have assessed the ability of ginger to alleviate nausea and vomiting, but it also has multiple effects on immune functioning. Ginger contains a vast amount of antioxidant compounds, nearly 40, which can be used to treat various inflammatory conditions [32]. Thus, the gingerols, shogaols, and diarylheptanoids in ginger may alleviate some symptoms from an inflammatory process. Some diseases, such as obesity, which is characterized by elevated levels of pro-inflammatory markers, can receive the benefit of being treated by ginger [33], especially through the paraoxonase-1 mechanism, which will avoid the deposit of lipidic compounds in vessel walls [22].

Some studies have described that, at the end of the treatment period with ginger, TNF-α levels decreased [34]; this illustrates that ginger may have a potential role as an adjuvant anti-inflammatory therapy for some conditions. Furthermore, ginger compounds such as 6-gingerol and 6-shogaol have an anti-inflammatory effect by inhibiting the production of inflammatory mediators, such as prostaglandin E2, NO, inflammatory cytokines (TNF-α), interleukin-1β (IL-1β)), and pro-inflammatory transcription factor (NF-κB). They also inhibit COX-1 and COX-2 [35]. Several authors have observed that 6-shogaol decreases nitric oxide (NO) synthesis more and inhibits arachidonic acid release to a greater extent than 6-gingerol [36,37].

## 4. Effect of Ginger on Inflammatory Diseases

In this context, numerous reports have pointed towards the fact that the bioactive compounds found in ginger can be effective in attenuating the symptoms of chronic inflammatory disorders [38].

### 4.1. Rheumatoid Arthritis

Rheumatoid arthritis (RA) is a chronic autoimmune disease characterized by peripheral pain involving joints (hands, feet, wrists, shoulders, elbows, hips, and knees). It is characterized by a tight interaction between cells and mediators of the innate and adaptive immune system [39]. While the cause of RA is unknown, it is influenced by genetic, epigenetic, and environmental factors. Pain, swelling secondary to inflammation of the synovial membrane, and stiffness occur, especially in the morning or after prolonged periods of rest. In addition to damaging the joints and surrounding tissues (tendons and muscles) that can lead to decreased mobility and joint function, chronic inflammation can affect other organs, such as the heart, lung, or kidney [40]. In addition, if the inflammation is high and sustained, it can cause fever, fatigue, asthenia, weight loss, and loss of appetite. Therefore, rheumatoid arthritis is considered a systemic disease [41].

The pathophysiological mechanisms for RA are not fully elucidated; however, it is clear that the immune system is compromised, causing chronic inflammation, and there is also oxidative stress [42,43]. Actual treatment for RA consists of immunomodulation or immunosuppression and symptomatic treatment (anti-inflammatories), but this treatment does not cure the disease. Since the development of RA depends on environmental factors, including diet, it is logical to think that certain foods or nutrients with an anti-inflammatory and antioxidant character may be useful in this pathology [44].

In this sense, ginger is a very good candidate, as it has antioxidant and anti-inflammatory properties. In fact, since ancient times, ginger has been used in medicine as an anti-inflammatory. It is currently known that this is due to the fact that the bioactive compounds in ginger are capable of inhibiting the COX-2 and LOX pathway [45]. Both in vitro and in vivo models have proved that ginger has antiarthritic effects [45,46].

One of the most abundant bioactive compounds of ginger is 6-shogaol. In vitro and in vivo, 6-shogaol has been shown to exhibit cancer protective effects, anti-inflammatory, antioxidant, and neuroprotective actions [47]. In vivo, 6-shogaol successfully reduced the formation of paw edema [48], leukocyte infiltration into the tissue, or symptoms of arthritis [49,50,51]. Bashir et al. verified how zingerone is capable of improving inflammation and oxidative stress in an animal model of arthritis [52].

Jo et al. [53], in a recent study, obtained very interesting results where 8-shoagol acts as a potent molecule against synovitis, showing a significant inhibitory effect against TNF-α-, IL-1β-, and IL-17-mediated inflammation and migration in an RA patient and 3D synovial culture system. Moreover, they proved that treatment with 8-shogaol reduced paw thickness and improved walking performance in the adjuvant-induced arthritic rat model, and reversed pathologies of joint structure in these rats and decreased inflammatory biomarkers in the joints [53].

In the recent study of Aryeian et al., it was observed that the effect of supplementation with 1.5 g/day of ginger in 63 patients with RA obtained a significant reduction in IL-1B and hs-CRP [54]. In addition, a significant decrease in TNF-α was observed in the ginger group.

### 4.2. Inflammatory Bowel Disease

Inflammatory bowel disease (IBD) is an umbrella term for various chronic inflammatory digestive processes, the etiology of which is unknown. Crohn’s disease and ulcerative colitis are among these processes. Crohn’s disease can be located anywhere in the gastrointestinal tract, from the mouth to the anus, although it usually occurs in the distal ileum and ascending colon. Ulcerative colitis is mainly located in the rectum and distal colon, although it may extend to the entire colon. The inflammatory process is discontinuous and asymmetrical. Unlike Crohn’s disease, ulcerative colitis starts in the rectum, where it gives rise to the so-called ulcerative proctitis [55]. The most common clinical symptoms of ulcerative colitis include abdominal pain, diarrhea, and bloody mucoid stools.

In relation to the pathogenesis of the disease, it is recognized that in addition to genetic factors, there are additional triggering factors, so-called environmental precipitants, and disease cofactors. In this sense, a precipitating factor would be an infection by a certain pathogen, coupled with a defective function of the intestinal barrier, which would trigger a chronic inflammatory response in genetically predisposed individuals with impaired regulation of the immune system.

The immune response of the gastrointestinal tract can lead to inflammation when the immune system and commensal bacteria balance is altered. This alteration occurs more frequently in genetically predisposed individuals. The gut microbiome plays an important role in the treatment of IBD [56]. For the already reported dysbiosis in patients with ulcerative colitis and Crohn’s disease compared to controls, Guo et al. [57] showed that ginger can improve the functions of the gut microbiota, restoring its diversity. Patients with Crohn’s disease and ulcerative colitis have an increased proportion of Proteobacteria and a decreased proportion of Firmicutes [58].

In gut homeostasis, Toll-like receptors (TLRs) play a pivotal role as mediators between the gut microbiome and the immune response. Pathogen-associated molecular patterns (PAMPs) are recognized by TLRs and upon recognition are activated, regulating dendritic cell maturation, adaptive immunity, and innate immunity. Expression of TLRs increases when these regulations are altered, leading to an increase in inflammatory cytokines and an increased risk of inflammatory bowel diseases [59]. TLR4 signaling is involved in inflammatory bowel disease and drug treatment efficacy [60].

Pharmacological treatment of IBD is based on non-specific anti-inflammatory and immunosuppressive drugs with suboptimal results. The goal of IBD therapy is to induce and maintain remission and ameliorate the disease’s secondary effects [61]. In addition to pharmacological treatment, diet plays a key role [62].

There are bioactive substances from different foods that can be used in the prevention and treatment of inflammatory bowel diseases. Among these substances is glycomacropeptide (GMP), a peptide derived from milk K-casein with immunomodulatory, bactericidal, and prebiotic effects. Curcumin is a yellow pigment from turmeric (Curcuma longa), a spice of Indian origin that has beneficial health effects. Curcumin ameliorates altered intestinal barrier function caused by inflammation by reducing myosin light chain kinase (MLCK) expression. It also decreases the production of TNF-α, IL-4, IL-6, and IL-13 by MCs in response to allergens [63].

There is a need to develop novel effective therapies and adjuvants for the treatment of IBD, as conventional treatment can have side effects (such as nausea, pancreatitis, allergic reactions) and is relatively often ineffective in some patients [64]. Ginger has antioxidant, antitumor, anti-inflammatory, and anti-ulcer effects, and has also been used for many years throughout the world to treat vomiting, diarrhea, and infections [65].

In IBD, ginger modulates the inflammatory response through suppression of nuclear factor kappa B (NF-κB), TNF-α, Nod-like receptor family proteins (NLRP), TLR, signal transducer of activators of transcription (STAT), mitogen-activated protein kinase (MAPK), and mTOR pathways, as well as inhibiting several proinflammatory cytokines (I L-6, IL-1β) and myeloperoxidase enzyme (MPO) [66,67].

Several authors have observed that gingerols and shogaols present in the rhizome decrease hepatic markers of inflammation by inhibiting NF-κB activity after consumption of a high-fat diet [68]. TNF-alpha inhibition leads to modulation of the inflammatory response, resulting in downregulation of the NF-κB signaling [69].

The oral consumption of gingerol, a bioactive substance in ginger, decreased the values of cytokines (IL-1beta, IL-6), TNF-alpha, NF-kB (p65), and increased IL-10 in an animal model of mice with ulcerative colitis induced by dextran sulfate sodium. Moreover, it decreased cyclooxygenase-2 (COX-2) enzyme activity and monocyte chemoattractant protein-1 (MCP-1) [66].

6-shogaol is a bioactive substance from dried ginger with anti-inflammatory properties. Several authors have observed that it increases the expression of the nuclear factor (erythroid-derived 2)-like 2 (Nrf-2) and heme oxygenase (HO-1) in DDS-induced colitis in a mouse model. It has also been shown to reduce the expression of IL-6, IL-1beta, and TNF-alpha [69,70]. However, the results are still scarce and in some cases contradictory, since in a pilot study of 45 patients with IBD, Van Tilburg et al. [71] observed that ginger did not have any beneficial effect when compared to a placebo group.

### 4.3. Systemic Lupus Erythematosus

Systemic lupus erythematosus (SLE) is a chronic autoimmune inflammatory connective tissue disease that affects the joints, kidneys, skin, mucous membranes, and blood vessel walls. SLE is characterized by overactivation of the autoimmune system with abnormal functions of innate and adaptive immune cells and the production of a large number of autoantibodies against nuclear components [72]. However, at the present time, there is no clear etiology for this inflammatory disease, although enough evidence is gathered about the implication of genetic and environmental factors. Additionally, a wide variety of clinical manifestations, comorbid conditions, and a complex pathogenesis coexist in SLE. According to the literature, for the disease to manifest, there must be (in addition to a genetic predisposition) different hormonal, nutritional, and environmental factors that contribute to its pathogenesis [73].

There are many immune cells and proinflammatory proteins that are part of the complex pathogenesis of lupus which could be therapeutic targets [74]. The pharmacological treatment of SLE is based on four main types of drugs: non-steroidal anti-inflammatory drugs, antimalarial drugs, anti-inflammatory corticosteroids, and biological therapies [75]. The right pharmacological treatment for SLE is still a matter of controversy, and multiple simultaneous approaches are currently being used. However, in recent years, diet, microbiota, stress, and physical activity are receiving more attention from researchers. Bearing in mind that in SLE there is a there is a dysfunction of the immune system and an exacerbated inflammatory response, those bioactive substances that improve immune function and reduce chronic inflammation could be useful in the treatment of this pathology. The bioactive compounds of ginger have been proved these actions. It is clear that ginger has a potent antioxidant and anti-inflammatory activity. Hence, the anti-inflammatory effects of ginger are yet to be further investigated in the context of SLE.

Currently, a recent study by Ali et al. [76] demonstrated a protective role for ginger-derived compounds in the context of SLE. Antiphospholipid syndrome (APS) is a comorbid condition which has a high prevalence in patients with lupus. Furthermore, APS is closely related to thrombus inflammatory disease, defined by the presence of circulating antiphospholipid antibodies. In this study, it is shown that 6-gingerol attenuates neutrophil extracellular trap release in response to SLE and APS-relevant stimuli through a mechanism that is at least partially dependent on phosphodiesterase inhibition.

### 4.4. Psoriasis

Psoriasis is a chronic inflammatory skin condition marked by keratinocyte overgrowth and inflammation, which leads to epidermal hyperplasia, a characteristic of lesioned psoriatic skin. The elbows, knees, and scalp are the most common sites for psoriatic plaques. There is still no therapy for psoriasis, despite recent research revealing aspects of the pathogenesis and the extensive interplay involving nerves, immune system, endocrine system, and skin cells [77]. Oxidative stress is a key factor in the development and progression of psoriasis, which is known to be caused by a number of factors, including alcohol consumption, smoking, infection, drugs, obesity, cell metabolism, immune response, and pathological state [78]. The production of reactive oxygen species is a critical step in the creation of oxidative stress in psoriasis.

They generally act as second messengers during this process and lead to an increase in the levels of oxidative products which result in the activation of Th1 and Th17 cells and keratinocytes through the MAPK, NF-kβ, and JAK-STAT pathways. This results in a cascade of inflammatory cytokines and growth factors. NF-kβ is an essential inflammatory mediator in the pathogenesis of psoriasis; increased expression of NF- β has been demonstrated in psoriatic lesions [79]. The phosphorylation of the inhibitor of kappa B kinase (IKK) complex by ROS can activate NF-kβ [80]. H_2_O_2_, which is transported by AQP3, has been linked to the activation of the NF-kβ signaling pathway in keratinocytes and the pathogenesis of psoriasis [81]. Altered NF-kβ signaling disrupts the balance of apoptotic signals, leading to the upregulation of cyclins and survivins, thereby inhibiting apoptosis. Furthermore, NF-kβ stimulates the synthesis of IL-17 and TNF-, boosting the inflammatory response downstream [78]. Eukaryotic cells use the NF-kβ pathway as a regulator of genes that affect cell proliferation and survival. NF-kβ regulates the inflammatory response by increasing the expression of inflammatory target genes such as cytokines, chemokines, and COX2. This enzyme increases the production of proinflammatory cytokines by triggering the creation of certain prostaglandins in response to inflammation.

Ginger inhibits inflammatory responses by decreasing NF-kβ, which results in a decrease in cytokine gene expression [82]. Several authors [27] have observed that administration for 21 days of ginger and metformin in liposomes decreases TNF-α and IL-22 levels. These results show that ginger’s bioactive compounds could be an alternative treatment for psoriasis treatment.

### 4.5. Cancer

There is growing evidence linking diet to cancer prevention and treatment. Cancer is the second leading cause of death in the world after cardiovascular disease, with important socio-economic consequences. Certain dietary components, such as ginger and its compound 6-gingerol, may be associated with a reduced risk of cancer development [83]. Inflammatory processes are associated with tumor progression [84]. Several authors [85] described the anti-inflammatory action of ginger extract in vivo cancer models in animals significantly reduced the elevated expression of TNF-alpha and NF-kβ in rats with liver cancer.

De Lima et al. [86] showed that ginger derivatives, as an extract or isolated compounds, exhibit relevant antiproliferative effects on tumoral cells, as well as anti-inflammatory activities. 6-gingerol is the most pharmacologically active compound possessing potential cancer protection properties via its effect on a variety of biological pathways involved in apoptosis, inhibition of angiogenesis, cell cycle regulation, and cytotoxic activity. Ginger extract and 6-gingerol exert their action through important mediators and pathways of cell signaling, including p38/MAPK, Bax/Bcl2, Nrf2, TNF-α, p65/NF-κB, p53, ERK1/2, SAPK/JNK, caspases-3/-9, and ROS/NF-κB/COX-2 [87].

Some cancer types such as breast neoplasms [88] have been associated with inflammation [33]. Thus, females that may experience two inflammatory processes (e.g., obesity and breast cancer) have been followed in clinical trials to assess ginger’s anti-inflammatory effects compared to placebo [89]. Ginger was associated with a decrease in IL-10 compared to the placebo group, and other advantages, such as reductions in insulin, glucose, insulin resistance, LDL-C, and triglycerides. Some benefits were also measured, such as increases in HDL-C and HDL-C/LDL-C compared to the initial moment in the study. Therefore, ginger supplementation successfully modulated both inflammatory and metabolic indicators, suggesting mitigation of inflammation symptoms. Since these biomarkers are connected to the inception of breast neoplasms, ginger supplementation may, therefore, exert some protection against cancer. Furthermore, patients in the early stages of cancer may also benefit from ginger supplementation, especially those patients receiving a chemotherapy treatment that causes a highly inflammatory internal process [23,90].

## 5. Conclusions

The present review shows how certain components present in ginger show efficacy against several inflammatory diseases. Compounds such as 6-shogaol, zingerone, and 8-shogaol show promising results in both human and animal models, decreasing some of the main symptoms of rheumatoid arthritis. 6-shogaol also appears to show a protective role against lupus by attenuating neutrophil extracellular traps through the inhibition of phosphodiesterases. Short-term administration of ginger could be an alternative treatment for psoriasis, since it is able to inhibit the inflammatory responses of this disease by decreasing NF-κB. Ginger supplementation could offer protection against cancer, especially in the early stages; it also seems to reduce the symptoms of aggressive treatments such as chemotherapy. Despite the reported findings, it seems necessary to keep on investigating the different anti-inflammatory actions of ginger to know their effects, as well as their possible synergies with other commercialized drugs. In IBD, ginger modulates the inflammation by the action of gingerols and shogaols; however, further research is needed, since results are contradictory. 6-gingerol demonstrated a protective role as a ginger-derived compound, since it attenuates neutrophil extracellular trap release in response to lupus by phosphodiesterase inhibition. Ginger inhibits inflammatory responses typical in psoriasis by decreasing NF-kβ, and its short-term administration shows that ginger may form part of an alternative treatment for psoriasis. Ginger may exert a function of supplementation and protection against cancer, especially for patients in the early stages of the disease, or for those receiving chemotherapy, reducing some of the symptoms of treatment. Further research is needed in the anti-inflammatory actions of ginger to further know its effects and potential synergies with marketed medicines.

## Figures and Tables

**Figure 1 molecules-27-07223-f001:**
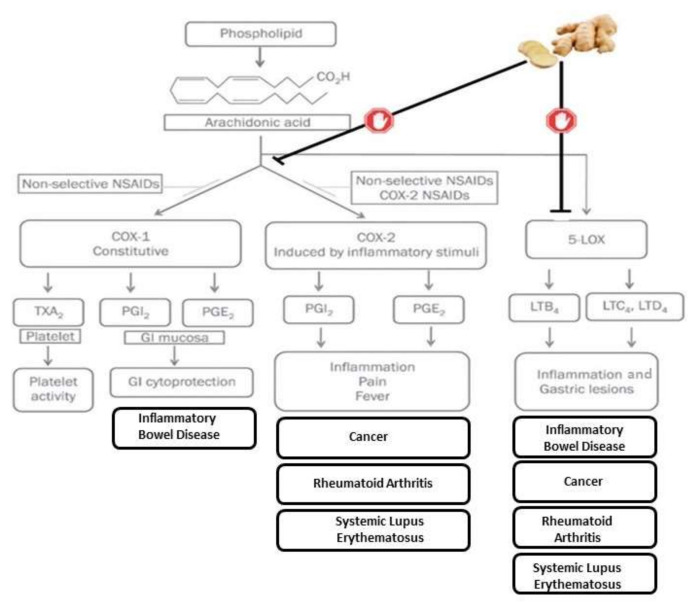
Inflammation, ginger actions, and its relationship with disease etiology and symptoms.

**Figure 2 molecules-27-07223-f002:**
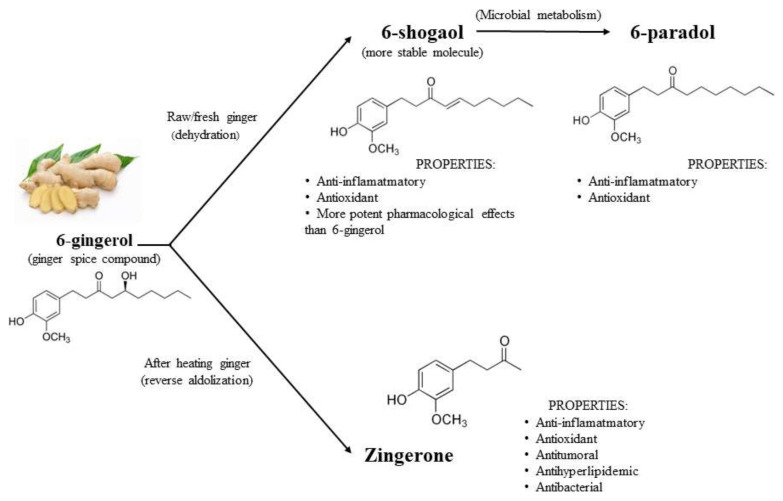
Properties and structure of the 4 main compound of ginger. Adapted of [3].

**Table 1 molecules-27-07223-t001:** Principal bioactives of the compounds of ginger.

Phenolic Compounds-Gingerol Analogues			Other Phenolic Compounds	Terpens
GINGEROLS	SHOGAOLS	PARADOLS		
6-gingerol 8-gingerol 10-gingerol 12-gingerol	6- shogaol 8-shogaol 10- shogaol dehydro-14-gingerdione 6-gingerdione 10-gingerdione	6-paradol 8-paradol 10-paradol	Zingerone Quercetin Gingerenone-A 6-dehydrogingerdione	β-bisabolene α-curcumene α-farnesene β-sesquiphellandrene Zingiberene

## Data Availability

Not applicable.

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
