# Peer review of "Effect of Ginger on Inflammatory Diseases"

_molecules, 2022, doi:10.3390/molecules27217223_

Round 1

Reviewer 1 Report

The study is focused on the potential of Ginger in the treatment of inflammatory chronic diseases through anti-inflammatory, immune-regulatory and antioxidative mechanisms and assessing the clinical trials in the respective area.

My main remarks on the manuscript are related to sections 2.2 and 4.1.5.

Section 2.2 L107-113 – the text is not dealing with kinetics, bioavailability and elimination, rather with stability and anti-inflammatory properties of series of biologically active compounds in ginger. The text should be reviewed and re-written.

Section 4.1.5. the claim that certain dietary compounds may prevent cancer  should not imply that ginger can treat or prevent cancer development, rather that the authors should use formulations like “may be associated with reduced risk of cancer development”.

I agree with the comments of the other reviewers, though I have considered them as technical discrepancies that I have not commented upon /e.g. low quality of Figure 1/. But in hindsight the accumulation of technical errors and/or discrepancies would ultimately lead to the necessity of a revision by the authors.

Author Response

The authors appreciate all the comments and suggestions of the reviewers, the indicated changes have been made, clearly contributing to the improvement of the article. We thank you in advance for the time you are going to dedicate to review the changes made. In this cover letter we explain, point by point, the details
of the revisions to the manuscript and our responses to the referees’
comments.

REVIEW 1

The study is focused on the potential of Ginger in the treatment of inflammatory chronic diseases through anti-inflammatory, immune-regulatory and antioxidative mechanisms and assessing the clinical trials in the respective area.

My main remarks on the manuscript are related to sections 2.2 and 4.1.5.

Section 2.2 L107-113 – the text is not dealing with kinetics, bioavailability and elimination, rather with stability and anti-inflammatory properties of series of biologically active compounds in ginger. The text should be reviewed and re-written.

This section has been modified to adapt the epigraph to its content. It has been rewritten and reorganized.

Section 4.1.5. the claim that certain dietary compounds may prevent cancer should not imply that ginger can treat or prevent cancer development, rather that the authors should use formulations like “may be associated with reduced risk of cancer development”.

All the section regarding to cancer has been re-read and modified as indicated.

I agree with the comments of the other reviewers, though I have considered them as technical discrepancies that I have not commented upon /e.g. low quality of Figure 1/. But in hindsight the accumulation of technical errors and/or discrepancies would ultimately lead to the necessity of a revision by the authors.

The quality of the figure 1 has been improved using IA well as the rest of the technical discrepancies (e.g., English has been reviewed and improved).

Reviewer 2 Report

Dear Authors,

 Your manuscript presents a study interesting and significant for its clinical aspects, even though the beneficial effects of ginseng ingredients are already widely reported in the scientific literature.

My recommendation is to accept manuscript for publication after major revision. For better understanding, I I have included my comments in the text of the manuscript, and I have listed some of them below:

MAJOR COMMENTS AND SUGGESTIONS

1/ The Abstract corresponds to the content of the work and contains the right keywords.

2/ In the literature, when describing plants, the names of the family, genus and species are always written in italics.

3/ Section “Mediators of the inflammatory process”. In this chapter, the antioxidant activity of ginseng ingredients should be emphasized. Indeed, free radicals trigger a chronic inflammatory reaction in the body, being one of the most important causes of tissue degeneration and the development of many diseases, but this is not the only harmful mechanism of their action.

There are also works documenting the influence of ginseng on the activity of another important antioxidant enzyme - paraoxonase 1, which is a component of the "third line of defense" against free radicals, e.g.

Carnuta MG, Deleanu M, Barbalata T, Toma L, Raileanu M, Sima AV, Stancu CS. Zingiber officinale extract administration diminishes steroyl-CoA desaturase gene expression and activity in hyperlipidemic hamster liver by reducing the oxidative and endoplasmic reticulum stress. Phytomedicine. 2018 Sep 15;48:62-69. doi: 10.1016/j.phymed.2018.04.059.

The authors could pay more attention to this enzyme, as they noted the antihyperlipidemic properties of ginseng. Paraoxonase is a calcium-dependent esterase located on HDL. It has been recently reported that antioxidant ability of HDL against LDL oxidation and accumulation of it in the vascular wall is largely due to the paraoxonase location on HDL.

4/ The same information is repeated in many places in the text, e.g. regarding anti-inflammatory or antioxidant activity, despite the fact that a special chapter devoted to them has been separated. This gives the work an appearance of chaos.

5/References:

1. Authors should carefully prepare the References section, as there are many inaccuracies in it.

2. The Authors should add more recent references to their list. Over 50% of items are older than 5 years.

3. Position 5 - wrong source and record. This is the correct quotation:

Mohd Yusof YA. Gingerol and Its Role in Chronic Diseases. Adv Exp Med Biol. 2016;929:177-207. doi: 10.1007/978-3-319-41342-6_8.

4. Position 69 - Why these data were not provided in accordance with Molecules requirements? just look at the previous, correctly written item - 68

5. There are no of page numbers – position 40, 45, 46, 47, 83

The formatting of the references is heterogeneous. I recommend preparing the references with a bibliography software package in line with Molecules requirements. Publisher asks for the number of the volume of the journal in which the article quoted was published. However there are no strict formatting requirements, the minor shortcomings in the formatting of the literature items exist:

- writing page numbering usually in full digits, but there are entries other than that

- why 28 items of the references have the abbreviation of the month and the rest are not?

- why some of the entries have the volume and issue numbers, and others  - the volume only (according to the requirements)

 6/Typographical errors:

- 4.1.2. Enfermedad inflamatoria intestinal (why in Spanish?)

- Chron's disease (line 212)

- Kbsignaling (line 250)

- missing dots, e.g.  lines 177, 186, 250, 284, 290, 321.

Author Response

The authors appreciate all the comments and suggestions of the reviewers, the indicated changes have been made, clearly contributing to the improvement of the article. We thank you in advance for the time you are going to dedicate to review the changes made. In this cover letter we explain, point by point, the details
of the revisions to the manuscript and our responses to the referees’
comments.

REVIEW 2

Dear Authors,

 Your manuscript presents a study interesting and significant for its clinical aspects, even though the beneficial effects of ginseng ingredients are already widely reported in the scientific literature.

My recommendation is to accept manuscript for publication after major revision. For better understanding, I I have included my comments in the text of the manuscript, and I have listed some of them below:

MAJOR COMMENTS AND SUGGESTIONS

1/ The Abstract corresponds to the content of the work and contains the right keywords.

2/ In the literature, when describing plants, the names of the family, genus and species are always written in italics.

The nomenclature of the plants has been modified

3/ Section “Mediators of the inflammatory process”. In this chapter, the antioxidant activity of ginseng ingredients should be emphasized. Indeed, free radicals trigger a chronic inflammatory reaction in the body, being one of the most important causes of tissue degeneration and the development of many diseases, but this is not the only harmful mechanism of their action.

There are also works documenting the influence of ginseng on the activity of another important antioxidant enzyme - paraoxonase 1, which is a component of the "third line of defense" against free radicals, e.g.

Carnuta MG, Deleanu M, Barbalata T, Toma L, Raileanu M, Sima AV, Stancu CS. Zingiber officinale extract administration diminishes steroyl-CoA desaturase gene expression and activity in hyperlipidemic hamster liver by reducing the oxidative and endoplasmic reticulum stress. Phytomedicine. 2018 Sep 15;48:62-69. doi: 10.1016/j.phymed.2018.04.059.

The authors could pay more attention to this enzyme, as they noted the antihyperlipidemic properties of ginseng. Paraoxonase is a calcium-dependent esterase located on HDL. It has been recently reported that antioxidant ability of HDL against LDL oxidation and accumulation of it in the vascular wall is largely due to the paraoxonase location on HDL.

As indicated, we have included and emphasized a part of antioxidant activity.

4/ The same information is repeated in many places in the text, e.g. regarding anti-inflammatory or antioxidant activity, despite the fact that a special chapter devoted to them has been separated. This gives the work an appearance of chaos.

Information repetitions have been removed to avoid the appearance of chaos.

5/References:

  1. Authors should carefully prepare the References section, as there are many inaccuracies in it.

Errors in bibliographic references have been corrected.

  1. The Authors should add more recent references to their list. Over 50% of items are older than 5 years.

Some more actual and recent references have been added.

The bibliography is as follows:

Total 87

55 (between 2016-2022) 63.2%.

19 (between 2010-2016) 21.8%

12 (between 2000-2010) 13.8%

1 (<2000)

  1. Position 5 - wrong source and record. This is the correct quotation:

Mohd Yusof YA. Gingerol and Its Role in Chronic Diseases. Adv Exp Med Biol. 2016;929:177-207. doi: 10.1007/978-3-319-41342-6_8.

It has been corrected.

  1. Position 69 -Why these data were not provided in accordance with Molecules requirements? just look at the previous, correctly written item – 68

They have been corrected

  1. There are no of page numbers – position 40, 45, 46, 47, 83

The indicated pages have been added

The formatting of the references is heterogeneous. I recommend preparing the references with a bibliography software package in line with Molecules requirements. Publisher asks for the number of the volume of the journal in which the article quoted was published. However there are no strict formatting requirements, the minor shortcomings in the formatting of the literature items exist:

- writing page numbering usually in full digits, but there are entries other than that

- why 28 items of the references have the abbreviation of the month and the rest are not?

- why some of the entries have the volume and issue numbers, and others  - the volume only (according to the requirements)

The format of the references has been adapted to the standards of the journal. Have been corrected all the mistakes and all the references have been revised.

 6/Typographical errors:

- 4.1.2. Enfermedad inflamatoria intestinal (why in Spanish?)

- Chron's disease (line 212)

- Kbsignaling (line 250)

- missing dots, e.g.  lines 177, 186, 250, 284, 290, 321.

All typographical errors have been corrected

Reviewer 3 Report

1

Abstract

In the abstract, the background part is a little long, whereas the results and conclusions of the literature review are scarce.

Please correct this.

2

Figure 1 is poor in visibility; please replace it with one of higher resolution.

In addition, the introduction section could include a paragraph about inflammatory diseases.

3

Section 2. Pharmacological properties of ginger

The title does not exactly reflect the content of this section.

4

Line 58, 72

The same sentence appears twice:

‘It is composed of multiple bioactive compounds that contribute to its recognized biological activities’

Please delete one of them.

5

Section 2.1. Bioactive compounds

Line 72–90

The materials from different references should be organized organically so as to keep the main line clear.

For example, they could be introduced in the sequence of: lipids, carbohydrates, terpenes, phenolic compounds, etc.

Line 79 describes the phenols.

‘Of the 400 types of compounds present in ginger….phenolic compounds’

The following sentence repeatedly state this class 'phenolic compounds.'

'Ginger is abundant in active constituents, such as phenolic and terpene compounds. (Line 84)'

When speaking the chemical compounds, I suggest the authors could provide a summary figure of the chemical structures of typical compounds of different chemical classes (may be placed in section 2.1. Bioactive compounds).

Line 91-96: ‘Pure 6-gingerol can be obtained by to….’

This paragraph is disordered. Please re-organize it.

Overall, the authors should give serious consideration to the organization of the reference materials, try to re-organize the manuscript, and keep the main line clear.

6

Line 140

The subsection title ‘4.1. Inflammatory diseases’ is dispensable. It can be deleted.

And the following subsections can be arranged like this:

4.1 Rheumatoid arthritis

4.2…

4.3…

4.4…

For each kind of disease, the introduction to the disease itself seems a little more superfluous. The description about the disease should be concise and keep the point to the effects of ginger to this disease. On the other hand, part of them (knowledge on the inflammatory diseases) might be summarized and presented in the ‘1. Introduction ’ section.

Besides, this part is about ‘Clinical trials using assessing ginger anti-inflammatory properties’, and therefore the review in this section could focus on the clinical trials of ginger products in various diseases, rather than fundamental research work involving the mechanisms and signaling pathways.

On the other hand, the reports on the mechanisms and signaling pathways that ginger involves could be stated mainly in the Section ‘3. Mediators of the inflammatory process’.

7

Line 260: ‘Authors (Van Tilburg et al.) observed in a pilot study of 45 IBS patients that ginger’

Delete the word ‘authors’.

Line 330: ‘De Lima et al. showed that ginger derivatives, in the form of an extract or isolated…(80)’

The citation should be placed immediately after the word ‘et al.’:

De Lima et al. (80) showed that ginger derivatives, in the form of an extract or isolated…

8

The conclusion should be presented in one paragraph that is organically organized.

The listing points are not like a typical conclusion.

9

As a review article, the reference materials should be organized in an organic way. However, this review manuscript is rather disordered, and is poorly organized.

In my opinion, it may do not fulfill the criterion of the Journal Molecules, unless the authors make significant revisions and improvements.

Author Response

The authors appreciate all the comments and suggestions of the reviewers, the indicated changes have been made, clearly contributing to the improvement of the article. We thank you in advance for the time you are going to dedicate to review the changes made. In this cover letter we explain, point by point, the details
of the revisions to the manuscript and our responses to the referees’
comments.

REVIEW 3

Abstract

In the abstract, the background part is a little long, whereas the results and conclusions of the literature review are scarce.

Please correct this.

The abstract has been modified according to the indications given.

  1. Figure 1 is poor in visibility; please replace it with one of higher resolution.

 The resolution of figure 1 has been increased

In addition, the introduction section could include a paragraph about inflammatory diseases.

3

Section 2. Pharmacological properties of ginger

The title does not exactly reflect the content of this section.

 The section title has been modified to match the content.

4

Line 58, 72

The same sentence appears twice:

‘It is composed of multiple bioactive compounds that contribute to its recognized biological activities’

Please delete one of them.

We have deleted line 72.

5

Section 2.1. Bioactive compounds

 Line 72–90

The materials from different references should be organized organically so as to keep the main line clear.

For example, they could be introduced in the sequence of: lipids, carbohydrates, terpenes, phenolic compounds, etc.

Line 79 describes the phenols.

‘Of the 400 types of compounds present in ginger….phenolic compounds’

The following sentence repeatedly state this class 'phenolic compounds.'

'Ginger is abundant in active constituents, such as phenolic and terpene compounds. (Line 84)'

The section has been reorganized to make it more ordered and clearer.

When speaking the chemical compounds, I suggest the authors could provide a summary figure of the chemical structures of typical compounds of different chemical classes (may be placed in section 2.1. Bioactive compounds).

A table have been introduced.

Line 91-96: ‘Pure 6-gingerol can be obtained by to….’

This paragraph is disordered. Please re-organize it.

 This paragraph has been reorganized.

Overall, the authors should give serious consideration to the organization of the reference materials, try to re-organize the manuscript, and keep the main line clear.

 The authors have revised and reorganized the different sections of the manuscript.

6

Line 140

The subsection title ‘4.1. Inflammatory diseases’ is dispensable. It can be deleted.

And the following subsections can be arranged like this:

4.1 Rheumatoid arthritis

4.2…

4.3…

4.4…

 The modification indicated by the reviewer has been made.

For each kind of disease, the introduction to the disease itself seems a little more superfluous. The description about the disease should be concise and keep the point to the effects of ginger to this disease. On the other hand, part of them (knowledge on the inflammatory diseases) might be summarized and presented in the ‘1. Introduction ’ section.

 Besides, this part is about ‘Clinical trials using assessing ginger anti-inflammatory properties’, and therefore the review in this section could focus on the clinical trials of ginger products in various diseases, rather than fundamental research work involving the mechanisms and signaling pathways.

On the other hand, the reports on the mechanisms and signaling pathways that ginger involves could be stated mainly in the Section ‘3. Mediators of the inflammatory process’.

The title of section 4 has been modified since, thanks to the reviewer's comments, we considered that it was not appropriate for what was described in the section. We have added some content in the introduction section about the diseases, however, since it is a review about ginger, we don’t want to provide much information about the conditions in the introduction section.

7

Line 260: ‘Authors (Van Tilburg et al.) observed in a pilot study of 45 IBS patients that ginger’

Delete the word ‘authors’.

Line 330: ‘De Lima et al. showed that ginger derivatives, in the form of an extract or isolated…(80)’

The citation should be placed immediately after the word ‘et al.’:

De Lima et al. (80) showed that ginger derivatives, in the form of an extract or isolated…

 All mistakes indicated by the reviewer have been corrected.

8

The conclusion should be presented in one paragraph that is organically organized.

The listing points are not like a typical conclusion.

The conclusions have been modified. They are presented as a paragraph.

9

As a review article, the reference materials should be organized in an organic way. However, this review manuscript is rather disordered, and is poorly organized.

In my opinion, it may do not fulfill the criterion of the Journal Molecules, unless the authors make significant revisions and improvements.

Following the reviewers' comments, the entire article has been revised and organized according to the contributions made. Duplications have been eliminated, the contents have been ordered, the headings have been adjusted to the content of the article and the bibliography and English have been revised.

Round 2

Reviewer 2 Report

Dear Authors,

Your article has almost been corrected, but you should also pay attention to the duplicate item 11 of the reference (the same is in item 15), because this changes the numbering throughout the article. Similarly, the proper notation of item 5 is: Mohd Yusof YA. Gingerol and Its Role in Chronic Diseases. Adv Exp Med Biol. 2016; 929: 177-207.

I have the impression that you do not pay attention to "cosmetic" corrections, which also affects the way your work is perceived. I am enclosing a document with the second correction.

Author Response

Reviewer 2

Dear Authors,

Your article has almost been corrected, but you should also pay attention to the duplicate item 11 of the reference (the same is in item 15), because this changes the numbering throughout the article. Similarly, the proper notation of item 5 is: Mohd Yusof YA. Gingerol and Its Role in Chronic Diseases. Adv Exp Med Biol. 2016; 929: 177-207.

Firstly, thank you for your comments that help to improve the quality of this revision.

According to your indications, reference 5 has been corrected and reference 11 and 15 have been unified.

I have the impression that you do not pay attention to "cosmetic" corrections, which also affects the way your work is perceived. I am enclosing a document with the second correction.

All "cosmetic" errors noted have been reviewed. We regret that they were not corrected earlier, as this is also important to us. We apologize for the lack of attention to this point.

Reviewer 3 Report

Dear Authors,

It looks like the manuscript has been re-organized and revised significantly.

Please refer to my previous comment:

When speaking the chemical compounds, I suggest the authors could provide a summary figure of the chemical structures of typical compounds of different chemical classes (may be placed in section 2.1. Bioactive compounds).

The authors should add a figure about the chemical structure of the major compounds of ginger.

Thank you.

Author Response

Reviewer 3

Dear Authors,

 It looks like the manuscript has been re-organized and revised significantly.

Please refer to my previous comment:

When speaking the chemical compounds, I suggest the authors could provide a summary figure of the chemical structures of typical compounds of different chemical classes (may be placed in section 2.1. Bioactive compounds).

The authors should add a figure about the chemical structure of the major compounds of ginger.

Thank you.

We appreciate your comments, and according to your suggestion we have considered adding a figure with the main structures and properties of the main compounds of ginger (figure 2).
